# Comparison of Two DNA Labeling Dyes Commonly Used to Detect Metabolically Active Bacteria

**DOI:** 10.3390/microorganisms13051015

**Published:** 2025-04-28

**Authors:** Leena Malayil, Suhana Chattopadhyay, Neha Sripathi, Emmanuel F. Mongodin, Amy R. Sapkota

**Affiliations:** 1Department of Global, Environmental, and Occupational Health, University of Maryland School of Public Health, College Park, MD 20742, USA; suhanac@umd.edu (S.C.); nsripath@umd.edu (N.S.); ars@umd.edu (A.R.S.); 2Institute for Genome Sciences, University of Maryland School of Medicine, Baltimore, MD 21201, USA; emmanuel.mongodin@nih.gov

**Keywords:** metabolically active bacteria, DNA labeling, next-generation sequencing, harvested rainwater, tobacco products, 5-bromo-2′-deoxyuridine, propidium monoazide

## Abstract

Bacteria are ubiquitous in the environment and critical to human health and disease, yet only a small fraction can be identified through standard culture methods. Advances in next-generation sequencing techniques have improved bacterial identification, but these DNA-based methods cannot distinguish live bacteria from relic DNA. Recently, DNA-labeling dyes (e.g., 5-bromo-2′-deoxyuridine [BrdU] and propidium monoazide [PMA]) have been used to detect metabolically active bacteria in different sample types. Here, we compare BrdU and PMA in combination with 16SrRNA gene sequencing to characterize metabolically active bacteria in two different sample types: (1) manufactured products (*n* = 78; cigarettes, hookah, and little cigar) and (2) natural samples (*n* = 186; rainwater, soil, and produce). Metabolically active bacterial communities identified in BrdU-labeled samples had lower alpha diversity than that of PMA-treated and non-treated samples. *Pseudomonas*, *Sphingomonas*, *Enterobacter*, and *Acinetobacter* were observed in all the samples tested. Irrespective of sample type, *Pseudomonas* was predominant in BrdU-treated samples, while *Acinetobacter* was more abundant in non-treated samples compared to PMA-treated samples. We also observed that PMA-treated samples tend to overestimate the metabolically active bacterial fraction compared to BrdU-treated samples. Overall, our study highlights how different labeling techniques influence bacterial community analysis findings, underscoring the need for careful selection of labeling approaches when assessing environmental samples.

## 1. Introduction

Traditional culture-based assays have long served as gold standards for detecting bacteria in diverse environmental sources. However, these techniques often are time-consuming and cannot identify non-culturable bacteria [1,2]. In contrast, molecular approaches such as polymerase chain reaction (PCR), quantitative PCR (qPCR), and multiplex PCR enable the quick and efficient detection of both culturable and non-culturable bacteria [3], but are limited in capturing the overall bacterial diversity in tested samples. Recently, next-generation sequencing (NGS) techniques have gained prominence for characterizing total bacterial communities (culturable and non-culturable) in complex environmental media [4,5,6]. However, a drawback lies in the inability of NGS to distinguish between live, metabolically active bacterial communities and those represented by free relic DNA not present in viable cells [7].

Previous studies have revealed relatively high levels of dead or relic DNA in a number of sample types relevant to human health, including environmental samples such as soil and water [8,9] and manufactured products such as tobacco [10]. These high levels of relic DNA can potentially lead to a misrepresentation of the overall viable bacterial community present [8,11]. However, it is critically important to understand whether bacterial communities detected in tested samples are live or dead (inactive) in order to best understand microbial human health risks that may be associated with environmental or tobacco exposures. Some bacteria (including human pathogens) can switch between metabolically active and dormant states to protect themselves from environmental stressors [12]. When conditions are favorable, rapid growth of these dormant bacteria to levels that surpass an infectious dose may potentially cause adverse human health effects among exposed individuals. For example, the rapid growth of metabolically active foodborne pathogens in irrigation water systems poses a significant concern, as these microorganisms can be transferred from water sources to food crops, potentially causing foodborne outbreaks [13]. In addition, the replication of viable bacteria within the tobacco microbiome could introduce additional health risks to tobacco users, given that viable potential pathogens have been detected in the mainstream smoke of cigarettes [14].

DNA-labeling techniques such as 5-bromo-2′-deoxyuridine (BrdU) (a synthetic thymidine analog that incorporates into replicating DNA) [15] and propidium monoazide (PMA) (a photoreactive DNA-binding dye that can penetrate membrane-compromised cells and, upon photo-activation, bind to free DNA) [16] have previously been employed to identify metabolically active bacteria in multiple environmental samples [7,15,17,18,19]. However, it is unclear how these labeling methods compare with one another. Therefore, we compared BrdU-labeled and PMA-labeled sub-samples, followed by NGS, and compared their efficiencies in characterizing metabolically active DNA and free relic DNA from two different sample types: (1) manufactured products (cigarettes, hookah, and little cigars) and (2) natural samples (rooftop-harvested rainwater, irrigated soil, and produce). By analyzing both manufactured products and natural samples, we aimed to determine how these labeling techniques influence bacterial community analysis across very diverse samples and to highlight the importance of selecting appropriate methods for assessing metabolically active bacteria in different environmental studies.

## 2. Materials and Methods

### 2.1. Sample Types and Sample Collection

Two different sample types (manufactured and natural) were included. The manufactured products included three different types of commercially available tobacco products (traditional cigarettes [Marlboro Menthol Gold, Marlboro Red, Newport Menthol Box, Newport Menthol Gold], little cigars [Swisher Sweets Cherry], cigarillos [Swisher Sweets Original], and hookah [Fumari ambrosia, Fumari mint chocolate chill, Fumari white gummi bear, Alfakher two apple, Alfakher mint, and Alfakher watermelon]). These products were purchased in stores located in College Park, MD, USA. The natural samples included produce, soil, and irrigation water samples collected from a rooftop harvested rainwater (RHRW) site located in Frederick, MD, USA, that utilizes harvested rainwater to irrigate produce. Details of the sampling site and sampling approach were described previously [20]. Briefly, from June to August 2018, 600 mL grab samples of harvested rainwater (*n* = 36) were collected over three sampling dates and included samples from the first flush tanks, secondary tanks, municipal water (control), and ambient rain. Samples from secondary tanks and ambient rain were collected only in the month of July due to the lack of rainfall events during the other two sampling dates. Additionally, 20 g of soil samples (*n* = 90) and produce samples (chard leaves, *n* = 60) irrigated with this water were collected aseptically into separate sterile Whirl-Pak^®^ bags and stored in a freezer until DNA extraction. 

### 2.2. BrdU and PMA Treatment of Natural Samples

#### 2.2.1. BrdU Treatment 

40 μL of 100 mM BrdU was added to 200 mL of each water sample, while another 200 mL of each sample was not subjected to any treatments (no treatment). The BrdU-treated samples were incubated for 2 days in the dark at room temperature, and all samples were filtered through 0.2 µm filters (as described below). For the soil samples, 0.2 g of the soil was weighed out in a 2 mL lysing matrix tube and incubated with 26 μL of 7.69 mM BrdU in the dark at room temperature for 48 h. Meanwhile, for the produce samples, 200 mL of sterile water was added to the Whirl-Pak^®^ bags, and the samples were hand-massaged for 30 s. Similar to the irrigation water samples, 40 μL of 100 mM BrdU was added to the 200 mL of produce wash water and incubated under the same conditions as the water samples. After incubation, all the resulting wash water was transferred by pipette to the filtration setup (described below). 

#### 2.2.2. Water Sample and Produce Wash Filtration 

The BrdU-treated, no-treatment samples and a separate 200 mL (subjected to PMA treatment, described below) were filtered through 0.2 μm, 47 mm filters (Pall Corporation, Port Washington, NY, USA). The filters were then dissected aseptically into four quadrants, placed in lysing matrix B tubes (MP Biomedicals, Solon, OH, USA), and then stored at −80 °C until DNA extraction.

#### 2.2.3. PMA Treatment

For the PMA treatment, the filters from the water and produce wash samples were added to the lysing matrix B tubes, along with 3 μL of 50 μM PMA, as described previously [8]. The PMA-treated samples were then subjected to a 5 min dark cycle and then exposed to a 650 W halogen lamp placed 20 cm from the sample tubes for 5 min. For the soil samples, 3 μL of 50 μM PMA was added to 0.2 g of soil in 2 mL lysing matrix B tubes, then subjected to a 5 min dark cycle and exposed to a 650 W halogen lamp placed 20 cm from the sample tubes for 5 min. All the samples (water *n* = 36; produce wash *n* = 60; and soil *n* = 90) were then stored at −80 °C until DNA extraction. 

### 2.3. BrdU and PMA Treatment of Manufactured Samples

Similar to the soil samples, 0.2 g of the tobacco from each type of tobacco product was weighed out into lysing matrix B tubes in duplicate, as previously described [21]. The samples were then incubated with 26 μL of 7.69 mM BrdU and kept in the dark at room temperature for 48 h. Another 0.2 g of each tobacco product was treated with 3 μL of 50 μM PMA, followed by a 5 min dark cycle and then exposure to a 650 W halogen lamp placed 20 cm from the sample tubes for 5 min. All treated (BrdU and PMA) and control (no treatment) samples were stored at −80 °C until DNA extractions were completed.

### 2.4. Immunocapturing of BrdU Treated Samples 

Immunocapture and isolation of BrdU-labeled DNA were performed using a previously published protocol [15]. Briefly, HS DNA/α-BrdU antibody complex was added to extracted denatured DNA from samples and incubated in the dark at room temperature with acetylated bovine serum albumin (BSA) in phosphate-buffered saline (PBS) washed magnetic beads (Dynabeads, Dynal Inc., New Hyde Park, NY, USA, Invitrogen by Thermofisher Scientific, Waltham, MA, USA). After incubation, the samples were washed in 0.5 mL PBS-BSA, and the BrdU-containing DNA fraction was eluted by adding 1.7 mM BrdU (in PBS-BSA) and incubating for 1 h in the dark at room temperature.

### 2.5. DNA Extraction, 16S rRNA Gene Amplification, and Sequencing 

Total genomic DNA extractions were performed using protocols previously published by our group [21]. Briefly, 1 mL of 1X PBS was added to the filters and the soil and tobacco samples in the lysing matrix B tubes before incubation with two enzymatic cocktails containing lysozyme, mutanolysin, proteinase K, lysostaphin, and sodium dodecyl sulphate SDS), after which, the cells were mechanically lysed using an MP Biomedical FastPrep 24 (Santa Ana, CA, USA). The DNA was then purified using the Qiagen QIAmp DNA mini kit (Germantown, MA, USA) per the manufacturer’s protocol.

Extracted DNA was PCR amplified for the V3–V4 hypervariable region of the 16S rRNA gene using the universal primers 319F (ACTCCTACGGGAGGCAGCAG) and 806R (GGACTACHVGGGTWTCTAAT) and sequenced on an Illumina HiSeq2500 (Illumina, San Diego, CA, USA) using a method developed at the Institute for Genome Sciences and described previously [22].

### 2.6. 16S rRNA Gene Sequencing Analysis

Sequencing data were processed using PANDAseq [23] to assemble 16S rRNA paired-end reads, followed by de-multiplexing, barcode/primer trimming, and chimera detection via UCHIME in QIIME (v1.9.1) [24]. Quality-filtered sequences were clustered into Operational Taxonomic Units (OTUs) and assigned taxonomy using VSEARCH [25] with a confidence threshold of 0.97 against the Greengenes database. Downstream data analysis and visualization were conducted in RStudio (v1.1.423) using multiple R packages. Normalization, when needed, was performed using metagenomeSeq’s cumulative sum scaling (CSS) [26]. Alpha diversity was assessed with the Shannon index [27], while beta diversity was evaluated using Bray–Curtis dissimilarity and compared via ANOSIM (999 permutations).

When appropriate, data were normalized with metagenomeSeq’s cumulative sum scaling (CSS) [26] to account for uneven sampling depth. Prior to normalization, alpha diversity was measured using the Shannon diversity index [27]. Bray–Curtis dissimilarity was used for calculating beta diversity and was compared using an analysis of similarities (ANOSIM) on normalized data (999 permutations). 

PICRUSt2 was used to predict the abundances of KEGG orthologs from the 16S rRNA data. The KO table and associated metadata were imported into RStudio (v. 2022.12.0+353, R version 4.2.2) using readxl (v. 1.4.2) and a pseudocount of 1 was added to all KO abundances to avoid log function errors in DESeq2 analysis. Differential expression analysis was conducted using DESeq2 (v. 1.38.3), and pairwise comparisons were conducted between (1) BrdU and no-treatment samples; (2) BrdU and PMA samples; and (3) PMA and no-treatment samples for the water, soil, produce, and tobacco samples. Based on the number of significant KEGG orthologs determined by DESeq2, a p-adjusted value cutoff of 0.001 or 0.01 was chosen. All the significant KEGG orthologs for each comparison were then mapped to KEGG pathways using the KEGG Orthology Database [28]. In Excel (v. 16.70), pathways were matched to KEGG abundances, and duplicate pathways were averaged. The consolidated tables comparing pathways to predicted abundances were then imported to RStudio using readxl, and phyloseq (v. 1.42.0) was used to subset the samples of interest in the pairwise comparison being conducted. The results were then plotted on a heatmap using heatmap (v. 1.0.12).

Additionally, to identify differentially abundant bacterial taxa and their effect size, bacterial abundance profiles were analyzed using Linear Discriminant Analysis Effective Size (LEfSe), along with the Kruskal–Wallis sum-rank test [29]. A False Discovery Rate (FDR)-adjusted *p*-value threshold of <0.05 and an LDA threshold of >2.0 were employed for statistical significance.

## 3. Results

### 3.1. Sequencing Dataset 

A total of 186 natural samples (*n* = 36 water, *n* = 90 soil, and *n* = 60 produce wash) and 78 manufactured tobacco samples (*n* = 24 cigarette tobacco samples, *n* = 36 hookah samples, and *n* = 18 little cigar samples) were PCR-amplified for the 16S rRNA gene and sequenced. Table 1 describes the sequencing dataset details.

### 3.2. Diversity Measures Between Sample Types and Treatments 

Shannon diversity (an alpha diversity metric) was calculated using both rarefied (after downsampling each sample to 6474 reads) (Figure 1) and non-rarefied data by treatments for both sample types (manufactured: cigarette, hookah, and little cigar; and natural: produce, soil, and water). Since no differences were observed between the rarefied and non-rarefied analysis, we only present the alpha-diversity analysis performed on the rarefied dataset. Irrespective of treatment, all manufactured samples (tobacco products) had lower Shannon diversity (Figure 1a) when compared to that of natural products (water, soil, and produce) (Figure 1b). Across each sample type, the average (SD) Shannon diversity indexes for all BrdU-treated samples (cigarettes: 0.825 (±0.331 SD); hookah: 0.417 (±0.180 SD); little cigar: 1.172 (±1.068 SD); produce: 3.484 (±0.494 SD); soil: 3.973 (±0.469 SD); and water: 3.161 (±0.495 SD)) were lower compared to their no-treatment and PMA-treated counterparts. However, statistically significantly lower alpha diversity was observed in BrdU-treated samples compared to non-treated samples (*p* < 0.05) only for cigarettes, hookah, and soil, and in PMA-treated samples compared to non-treated samples (*p* < 0.05) only for cigarettes and hookah. Additionally, comparing across all sample types, non-treated and PMA-treated soil had the highest alpha diversity (no treatment 6.382 (±0.196 SD); PMA 5.989 (±1.088 SD)). 

Beta diversity between all normalized samples was computed using PCoA plots of Bray–Curtis dissimilarity (Figure 2) and showed significant clustering by treatment across all sample types (*p* < 0.05) except water (ANOSIM R: 0.0599, *p* = 0.087). Additionally, we observed that there were no significant differences between non-treated and PMA-treated samples within any of the tested samples (cigarette: ANOSIM R: 0.03439, *p* = 0.288; hookah: ANOSIM R: 0.06186, *p* = 0.102; little cigars: ANOSIM R: −0.07143, *p* = 0.66667; water: ANOSIM R: −0.0665, *p* = 0.94; soil: ANOSIM R: −0.0114, *p* = 0.731; and produce: ANOSIM R: −0.03709, *p* = 0.948).

### 3.3. Bacterial Taxonomic Variations Between Treatments and Across Sample Types

To evaluate the extent of variation between treatments, we utilized LEfSe analysis to discern significant taxonomic differences among samples treated with BrdU and PMA, and nontreated samples, across natural and manufactured sample types. Our analysis identified four common bacterial genera across both sample types: *Pseudomonas*, *Sphingomonas*, *Enterobacter*, and *Acinetobacter*. Furthermore, our observations revealed a higher relative abundance of *Sphingomonas* in the non-treated manufactured samples, whereas in the natural samples, *Sphingomonas* exhibited a higher relative abundance in the BrdU-treated samples. This trend was mirrored in the case of the *Enterobacter* genus. Conversely, regardless of sample type, *Pseudomonas* and *Acinetobacter* were found to be more abundant in BrdU-treated and non-treated samples, respectively, compared to samples treated with PMA.

Looking more closely at the BrdU-treated samples, we identified a significantly higher relative abundance of five bacterial taxa (*Pseudomonas*, *Terribacillus*, *Propionibacterium*, *Delftia*, and *Streptococcus*), four bacterial taxa (*Pseudomonas*, *Propionibacterium*, *Terribacillus*, and *Lactobacillus*), and five bacterial taxa (*Pseudomonas*, *Propionibacterium*, *Delftia*, *Terribacillus*, and *Stenotrophomonas*) within the cigarette (Figure 3a), hookah (Appendix A), and little cigar (Appendix A) samples, respectively, compared to the non-treated and PMA samples. Within the BrdU-treated natural samples, we identified 12 bacterial taxa (*Pseudomonas*, *Chryseobacterium*, *Enterobacter*, *Rhizobium*, *Microbacterium*, *Janthinobacterium*, *Arthrobacter*, *Aeromonas*, *Sphingomonas*, *Curvibacter*, *Exiguobacterium*, and *Stenotrophomonas*), 13 bacterial taxa (*Chryseobacterium*, *Enterobacter*, *Rhizobium*, *Exiguobacterium*, *Arthrobacter*, *Sphingomonas*, *Paenibacillus*, *Microbacterium*, *Pantoea*, *Curvibacter*, *Massilia*, *Bacillus*, and *Stenotrophomonas*), and 12 bacterial taxa (*Bradyrhizobium*, *Flavobacterium*, *Enterobacterium*, *Roseobacter*, *Pseudomonas*, *Sphingomonas*, *Rhizorhabdus*, *Acientobacter*, *Bryobacter*, *Methylobacterium*, *Lysobacter*, and *Pantoea*) in soil (Figure 3b), water (Appendix A), and produce samples (Appendix A), respectively, at a higher relative abundance when compared to PMA and non-treated soil samples.

### 3.4. Predictive Functional Profile of Bacterial Communities Between Treatments

Using the PICRUSt tool, we predicted functional profiles of the bacterial community based on the relative abundance of the 16S rRNA marker gene in our tested samples. Irrespective of sample types, a key observation was that BrdU-treated samples had lesser abundant functional genes when compared to the non-treated samples (Figure 4a and Figure 5a, Appendix A), while a comparison between PMA-treated and non-treated samples did not show a drastic difference in the relative abundance of functional genes (Figure 4b and Figure 5b, Appendix A).

## 4. Discussion

In this study, we compared the use of two DNA-labeling dyes (i.e., 5-bromo-2′- deoxyuridine [BrdU] and propidium monazide [PMA]) to detect metabolically active bacteria communities among manufactured and natural samples. These DNA-labeling dyes have been extensively used to characterize metabolically active bacterial communities from multiple samples [7,8,17,19,30]. However, to our knowledge, no previous studies have conducted a head-to-head comparison between the dyes used in tandem with next-generation sequencing techniques to understand if either one would be a better indicator (than the other) in identifying metabolically active bacterial communities within different samples. Hence, here, we compared the efficacy of the two DNA-labeling dyes (BrdU and PMA), coupled with 16S rRNA gene sequencing, to characterize bacterial communities across two overall sample types: (1) manufactured and (2) natural. Overall, we observed that BrdU-treated samples were characterized by significantly lower alpha diversity when compared to non-treated and PMA-treated samples irrespective of the tested sample types. Additionally, while a significantly higher bacterial diversity was observed among the PMA-treated manufactured samples (cigarette and hookah), this might indicate an overestimation of live bacterial communities, which corroborates an observation made previously by Li et al. (2017), while comparing DNA-, PMA-, and RNA-based 16S rRNA gene sequencing to detect bacterial communities in different water sources [7].

The selective targeting and elimination of dead cells during sample analysis approaches is an important aspect of many research studies and medical applications [31]. Among such approaches, two DNA-labeling dyes that are often compared are BrdU and PMA. While both dyes can be effective in identifying target organisms in certain situations, the key differences exist in how each of these dyes incorporates into DNA.

BrdU is a chemical that can be incorporated into replicating DNA during the process of cell division and, thus, has often been used to identify and study actively proliferating cells. By using an antibody that specifically binds to BrdU, researchers can selectively target and eliminate cells that have gone through cell division. This can be a useful way to identify and study actively proliferating cells. However, it is important to note that BrdU only labels cells that are actively dividing and does not label cells that have stopped dividing or are not actively proliferating (such as dormant cells or spores) [15]. Moreover, while cell-specific BrdU uptake efficiencies may vary, Hellman et al. [32] demonstrated no apparent correlation between specific taxa affiliation and its ability to incorporate BrdU into its DNA. Our findings revealed the presence of metabolically active Gram-negative (*Pseudomonas*, *Delftia*, *Chryseobacterium*, *Enterobacter*, *Rhizobium*, *Janthinobacterium*, *Arthrobacter*, *Aeromonas*, *Sphingomonas*, *Curvibacter*, and *Stenotrophomonas*) and Gram-positive (*Terribacillus*, *Propionibacterium*, *Streptococcus*, *Microbacterium*, and *Exiguobacterium*) bacterial communities in BrdU-treated samples, which corroborates the Hellman et al. (2011) study. Previous tobacco-related work from our group [10,14,21] and others [33,34] demonstrated the presence of viable *Pseudomonas* and *Terribacillus* in both tobacco products and cigarette smoke, two types of microorganisms that were also identified among the BrdU-treated tobacco samples in the present study. Yet, one major disadvantage of the BrdU technique is that it can be quite time-consuming (overnight incubation, plus immunocapturing of the BrdU samples) and expensive, which may limit its application in certain situations, especially during a foodborne outbreak or in other situations where rapid analysis is imperative.

In contrast to the process of incorporating BrdU in metabolically active cells, PMA selectively crosslinks with DNA in dead cells (relic DNA) in the presence of light. This crosslinking inhibits amplification of relic DNA during subsequent PCR analysis and helps reduce false positives in the data, making PMA particularly useful in identifying the living bacterial population in a sample [16]. Hence, using PMA as a DNA-labeling dye is thought to provide a more accurate representation of the living bacterial population, since it excludes dead cells and other extraneous material from the data [16]. Yet, some studies have shown the possibility of PMA methods generating false negatives, especially in samples with a high concentration of dead cells or those derived from biofilms. For example, Taylor et al. (2014) observed that PMA, coupled with qPCR, failed to detect live *Legionella* in biofilm samples [35]. A recent study conducted by Wang et al. (2021) demonstrated that the utilization of PMA alongside high-throughput sequencing techniques does not provide an accurate quantification of the viability of the entire microbial community, particularly in complex environments such as human saliva and soil [36]. These observations were corroborated through our PICRUST analysis (Figure 4 and Figure 5), which showed no difference in the relative abundance of functional genes between non-treated samples and PMA-treated samples. Another drawback associated with the use of PMA is that it has been observed that higher concentrations of PMA treatment can be increasingly toxic to bacteria, resulting in the death of these bacteria, which might result in an overestimation of dead bacteria in a given sample [7,35]. Aside from these disadvantages, the use of PMA is less time-consuming when compared to the use of BrdU and is thereby more attractive for the rapid detection of bacterial communities from various sources.

In conclusion, our data demonstrated lower alpha diversity across BrdU-treated samples when compared to that of PMA-treated and non-treated samples, indicating lower diversity in metabolically active bacteria in all the tested samples. Interestingly, PMA-treated samples tended to overestimate the metabolically active fraction of bacteria relative to BrdU-treated samples, suggesting that while PMA can potentially indicate viability, it may be less precise in distinguishing active bacteria in certain conditions. However, neither technique is without limitations, as mentioned above; each has unique strengths and weaknesses that can impact the interpretation of microbial community dynamics. Ultimately, the choice of labeling technique should be guided by the particular needs of the study or application, with careful consideration of the benefits and drawbacks of each option.

## Figures and Tables

**Figure 1 microorganisms-13-01015-f001:**
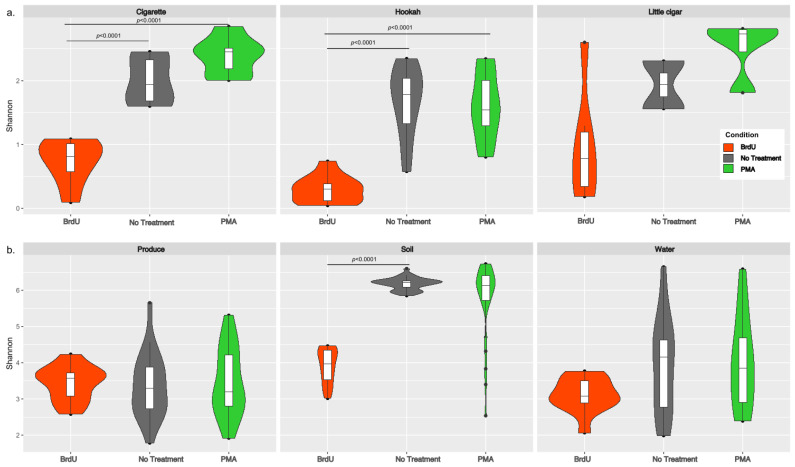
Violin plots of alpha diversity (Shannon diversity index) across (**a**) cigarette, hookah, and little cigars and (**b**) produce, soil, and water with treatments (BRDU and PMA) and without treatments on rarefied data to minimum sampling depth. The alpha diversity of BrdU-treated (orange) and PMA-treated (green) samples represents the diversity observed in the metabolically active fraction of bacterial communities present in each sample when compared to no-treatment (grey) samples.

**Figure 2 microorganisms-13-01015-f002:**
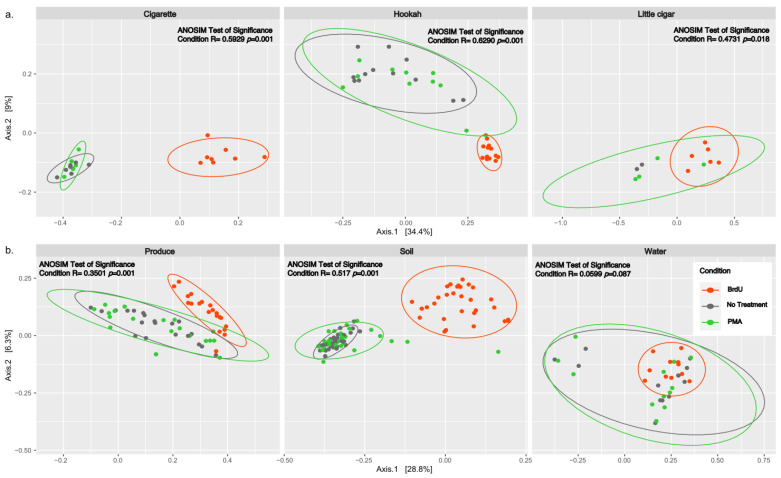
PCoA analysis of Bray–Curtis computed distances between treated (BrdU and PMA) and non-treated (noTRT) (**a**) cigarette, hookah, and little cigars and (**b**) produce, soil, and water. Orange depicts BrdU-treated samples, green depicts PMA-treated samples, and grey depicts non-treated samples. Solid-colored ellipses are drawn at 95% confidence intervals for treatments in each sample type.

**Figure 3 microorganisms-13-01015-f003:**
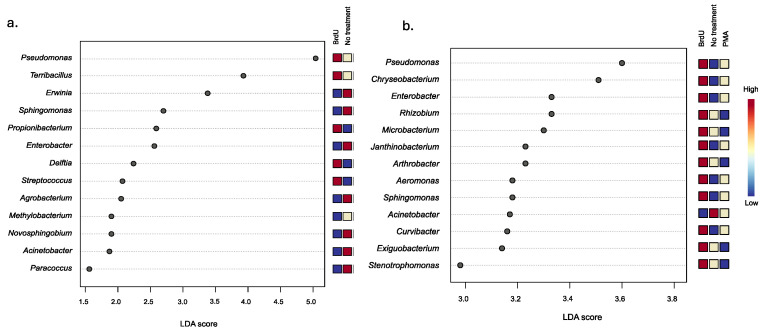
LEfSe analysis of the differential abundance of bacteria between treatments (BrdU, no-treatment, and PMA) in (**a**) manufactured (cigarettes) and (**b**) natural samples (soil).

**Figure 4 microorganisms-13-01015-f004:**
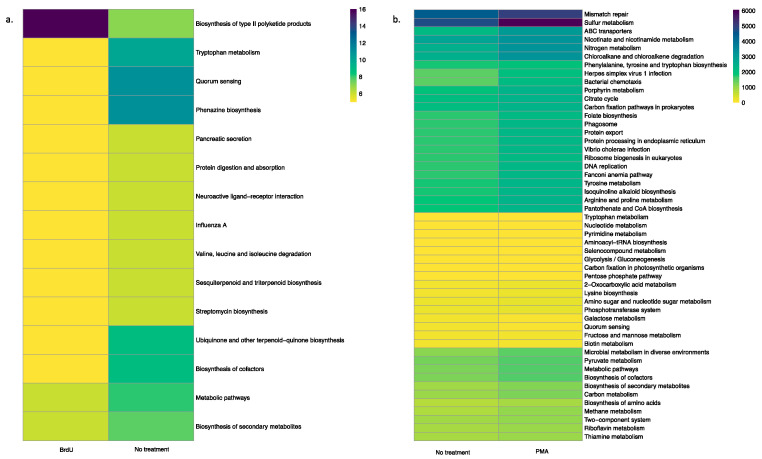
Heatmap illustrating the functional profile predicted at level 2 KEGGs Orthologs using PICRUSt analysis of a natural sample (water): (**a**) BrdU-treated versus no treatment and (**b**) PMA-treated versus no treatment.

**Figure 5 microorganisms-13-01015-f005:**
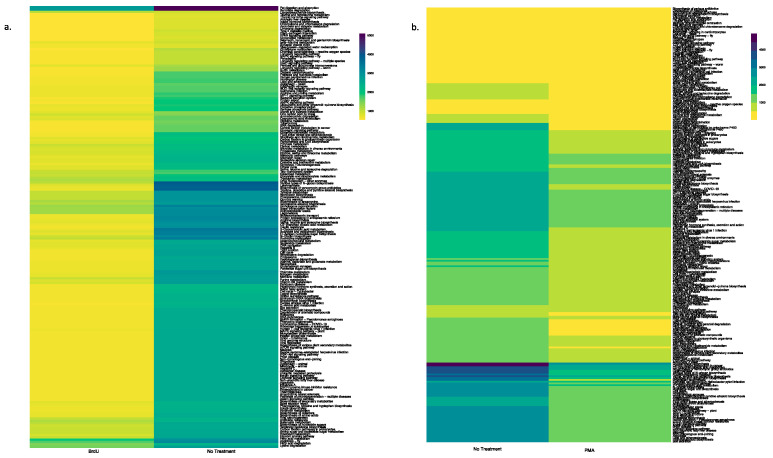
Heatmap illustrating the functional profile predicted at level 2 KEGGs Orthologs using PICRUSt analysis of a manufactured sample (hookah): (**a**) BrdU-treated versus no treatment and (**b**) PMA-treated versus no treatment.

**Table 1 microorganisms-13-01015-t001:** Sequencing details of the dataset.

	Natural Samples	Manufactured Samples
Initial number of samples	186 (*n* = 36 water, *n* = 90 soil, and *n* = 60 produce)	78 (*n* = 24 cigarette tobacco, *n* = 36 hookah, and *n* = 18 little cigar)
Total number of raw seq.	5,975,496	1,007,780
Initial number of OTUs	15,071	1216
Min number of reads	445	61
Max number of reads	62,676	261,593
Number of sequences per sample (average ± SD)	32,126.32 (±11,403.18 SD)	13,996.94 (±31,018.68 SD)
Good’s coverage cutoff	0.90	0.85
Final number of OTUs	8791	609
Final number of sequences	5,582,816	638,852
Final number of samples	184 (*n* = 36 water, *n* = 89 soil, and *n* = 59 produce)	68 (*n* = 21 cigarette tobacco, *n* = 35 hookah, and *n* = 12 little cigar)

## Data Availability

Data concerning the samples included in this study are deposited under two NCBI BioProject accession numbers: PRJNA473598 (synthetic samples) and PRJNA483768 (natural environmental samples).

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
