# Peer review of "Comparison of Two DNA Labeling Dyes Commonly Used to Detect Metabolically Active Bacteria"

_microorganisms, 2025, doi:10.3390/microorganisms13051015_

Round 1
Reviewer 1 Report
Comments and Suggestions for Authors
Dear authors,
After reading your article, I have some questions:
- The purpose of the work is missing, it must be added.
- Introduction section - all references are very old.
- It is not clear why samples of tobacco products were taken for experiments as manufactured products.
- Line 85 - “BrdU and PMA treatment of natural samples:” - The authors probably forgot to put it in bold.
- Lines 111-113 “Similar to the soil samples, 0.2g of the tobacco from each type of tobacco product was weighed out into lysing matrix B tubes in duplicate, as previously described [14–17].” - What is the point of listing all your works in which this method is described? In my opinion, one primary source is enough.
- Lines 119-120 “Immunocapture and isolation of BrdU- labeled DNA were performed using a previously published protocol [8,17,18]” – Leave a reference only to the original source.
- Line 109 “-80°C” – Please add a space between the numeric value and the degree symbol, check this throughout the text.
- Line 114 - “7.69mM”, Line 115 “ 50μM” – Please add a space between the numerical value and the units of measurement. Please correct this point throughout the manuscript.
- Lines 127-129 “Total genomic DNA extractions were performed using protocols previously published by our group [14– 18].” – It is sufficient to provide a reference to only one primary source.
- Figure 3 - it is not clear why only soil samples were analysed as natural samples? If the authors have data on water samples, why are these not shown?
- In Figures 4 and 5 it is not clear why not all samples were taken for this analysis, but only some (water and hookah)?
- Line 288 – “(Urbach et al.[8])” – the reference must be made in the format approved by the journal.
- Lines 295-296 “… the Hellman et al. (date) study.” - It is not clear what the word in brackets means, apparently the year of publication should be in brackets?
- It is not clear why the authors collected so many different types of samples if they only used some of them? Wouldn't it have been better to take 2-3 types of samples (with replicates) and do all the comparative analyses with them?
Reviewer 2 Report
Comments and Suggestions for Authors
The topic of the manuscript falls into the scope of Microorganisms Journal. It is interesting to compare the efficiency of different DNA-labeling dyes in terms of detection of metabolically active bacteria. The information of only active bacteria is important when one studies some metabolic aspects of them so it is crucial to focus only of living fraction of the microbes.
The title is informative giving clear message about the content of the manuscript. Abstract is ok and contain sufficient information summarizing text. Keywords are adequate to the topic. Introduction is concise and highlight the need of knowing differences between studied dyes in proper detection of living/active fraction of microorganisms.
The experiment was well designed and conducted, all key information are provided with proper references of the previous studies. There was very good bioinformatic tools used to provide genuine results. The result section is also well written accompanied by very good figures, all providing convincing information. Only Fig. 5 contains so many details that makes is hard to read. Data are also well discussed what make one believe that there are differences between both dyes what leads to clear conclusions - PMA is less precise.
For me this manuscript is a good example for those which are so good that may be published without any corrections.
Author Response
Thank you so much for appreciating our work and for the kind remarks
Round 2
Reviewer 1 Report
Comments and Suggestions for Authors
Dear authors, thank you for correcting your manuscript. However, there is still one global problem that, in my opinion, reduces the quality of the article - this is an unreasonable use in the study of different types of samples (natural and tobacco products). Some comments on this remark are presented below:
- Lines 64-66 - it is not clear how the two types of samples relate to each other. The choice between natural and industrial (tobacco products) samples needs to be justified. In this case, it would be more logical to use either only natural samples or only tobacco products, unless there is a strong justification for comparing these two types of samples.
- Lines 66-69 – “By analyzing manufactured products and natural samples, the study aims to determine how these labeling techniques influence bacterial community analysis and to highlight the importance of selecting appropriate methods for assessing metabolically active bacteria in environmental studies.” - The purpose of your paper refers only to the environment, but nothing is said about tobacco samples, which as far as I know are not part of the environment. There is also no mention of the need to carry out such studies on tobacco samples in the entire text of the introduction.
- Lines 71-72 – “Two different environmental sample types (manufactured and natural) were included” – Why do you think that manufactured samples are environmental samples?
- In the discussion section, the authors do not in any way compare the results obtained with natural samples and tobacco products. The authors discuss methods and dyes. From this we can conclude that the type of sample is not so important in this paper. And it would be more logical to keep the results only with, for example, water and soil samples.
